# VinDr-RibCXR: A Benchmark Dataset for Automatic Segmentation and Labeling of Individual Ribs on Chest X-rays

**Hoang C. Nguyen**[1]                                    canhhoang30011999@gmail.com
**Tung T. Le**[1]                                         lethanhtung.tung06@gmail.com
**Hieu H. Pham**[1,2]                                     v.hieuph4@vinbigdata.org
**Ha Q. Nguyen**[1,2]                                     v.hanq3@vinbigdata.org

[1] *Medical Imaging Department, Vingroup Big Data Institute, Hanoi, Vietnam*

[2] *College of Engineering and Computer Science, VinUniversity, Hanoi, Vietnam*

**Editors:** Under Review for MIDL 2021

## Abstract

We introduce a new benchmark dataset, namely VinDr-RibCXR, for automatic segmentation and labeling of individual ribs from chest X-ray (CXR) scans. The VinDr-RibCXR contains 245 CXRs with corresponding ground truth annotations provided by human experts. A set of state-of-the-art segmentation models are trained on 196 images from the VinDr-RibCXR to segment and label 20 individual ribs. Our best performing model obtains a Dice score of 0.834 (95% CI, 0.810–0.853) on an independent test set of 49 images. Our study, therefore, serves as a proof of concept and baseline performance for future research.

**Keywords:** Rib segmentation, CXR, Benchmark dataset, Deep learning.

## 1. Introduction

A wide range of diagnostic tasks can benefit from an automatic system that is able to segment and label individual ribs on chest X-ray (CXR) images. To this end, traditional approaches (Candemir et al., 2016) exploited hand-crafted features to identify the ribs, but failed with anterior ribs. Recently, deep learning (DL) has shown superior performance to other methods in the segmentation and labeling of individual ribs (Wessel et al., 2019). However, developing DL algorithms for this task requires annotated images for each rib structure at pixel-level. To the best of our knowledge, there exists no such benchmark datasets and protocols. Hence, we present VinDr-RibCXR – a benchmark dataset for the automatic segmentation and labeling of individual ribs on CXRs. This work also reports performance of several state-of-the-art DL-based segmentation models on the VinDr-RibCXR dataset. The dataset and codes will be made publicly available[1] to encourage new advances.

## 2. Material and Method

### 2.1. Dataset

We built a dataset, namely VinDr-RibCXR, of 245 AP/PA CXR images for segmentation and labeling of individual ribs. The raw images in DICOM format were sourced from VinDr-CXR dataset (Nguyen et al., 2020), for which all scans have been de-identified to protect patient privacy. We then designed an in-house web-based labeling tool called VinDr Lab (https://vindr.ai/vindr-lab) that allows to segment individual ribs at pixel-level on

---

1. https://github.com/vinbigdata-medical/MIDL2021-VinDr-RibCXR

DICOM scans. Each image was assigned to an expert, who manually segmented and annotated each of 20 ribs, denoted as L1→L10 (left ribs) and R1→R10 (right ribs). The masks of ribs (see Figure 1) were then stored in a JSON file that can later be used for training instance segmentation models. To the best of our knowledge, the VinDr-RibCXR is the first publicly released dataset that includes segmentation annotations of the individual ribs, and for both anterior and posterior ribs. To develop and evaluate segmentation algorithms, we divided the whole dataset into a training set of 196 images and a validation set of 49 images.

### 2.2. Model development

We deployed various state-of-the-art DL-based segmentation models for the task of individual rib segmentation and labeling, including U-Net (Ronneberger et al., 2015), U-Net with EfficientNet-B0 (Tan and Le, 2019), Feature Pyramid Network (FPN) (Lin et al., 2017) with EfficientNet-B0, and U-Net++ (Zhou et al., 2018) with EfficientNet-B0. These models are well-known to be effective for many medical image segmentation tasks. To train the networks, we resized all training images to 512×512 pixels. Dice loss and Adam optimizer were used to optimize the network's parameters. All models were trained for 200 epochs with a learning rate of $1e-3$. Some augmentation techniques (*i.e.* horizontal flip, shift, scale, rotate, and random brightness contrast) were applied to avoid overfitting. The models were implemented using Pytorch (v1.7.1) and trained on a NVIDIA RTX 2080 Ti GPU.

## 3. Evaluation and Result

The effectiveness of the segmentation models was evaluated by Dice score, 95% Hausdorff distance (95% HD), sensitivity, and specificity. Experimental results reported by the proposed segmentation models are shown in Table 1. Our best performing model, which was Nested U-Net with EfficientNet-B0 encoder, achieved a Dice score of 0.834 (95% CI, 0.810–0.853), a 95% HD of 15.453 (95% CI, 13.340–17.450), a sensitivity of 0.841 (95% CI, 0.812–0.858), and a specificity 0.998 (95% CI, 0.997–0.998). Figure 1 shows segmentation and labeling results of the Nested U-Net on some representative cases from the validation set of VinDr-RibCXR and external datasets including JSRT and Shenzen. We observe that the algorithm correctly segmented the individual ribs on both normal and abnormal CXR scans, for both the VinDr-RibCXR and two external datasets. Although the abnormal CXR contains ribs that are obscured by lung abnormalities, the proposed algorithm still produced a considerably accurate segmentation mask for all ribs.

Table 1: Segmentation performance on the validation set of VinDr-RibCXR.

| Model | Dice | 95% HD | Sensitivity | Specificity |
|---|---|---|---|---|
| U-Net | .765 (.737–.788) | 28.038 (22.449–34.604) | .773 (.738–.791) | .996 (.996–.997) |
| U-Net **w.** EfficientNet-B0 | .829 (.808–.847) | 16.807 (14.372–19.539) | .844 (.818–.858) | .998 (.997–.998) |
| FPN **w.** EfficientNet-B0 | .807 (.773–.824) | 15.049 (13.190–16.953) | .808 (.773–.828) | .997 (.997–.998) |
| U-Net++ **w.** EfficientNet-B0 | .834 (.810–.853) | 15.453 (13.340–17.450) | .841 (.812–.858) | .998 (.997–.998) |

## 4. Conclusion

In this work, we introduced a new benchmark dataset and baseline predictive models for automatically segmenting and labeling individual ribs from chest radiographs. Our experimental results have shown the effectiveness of the DL-based segmentation models for this task. This study serves as a proof of concept and baseline performance for future research.

Future works include investigating more powerful DL models and extending the dataset to enhance the segmentation performance.

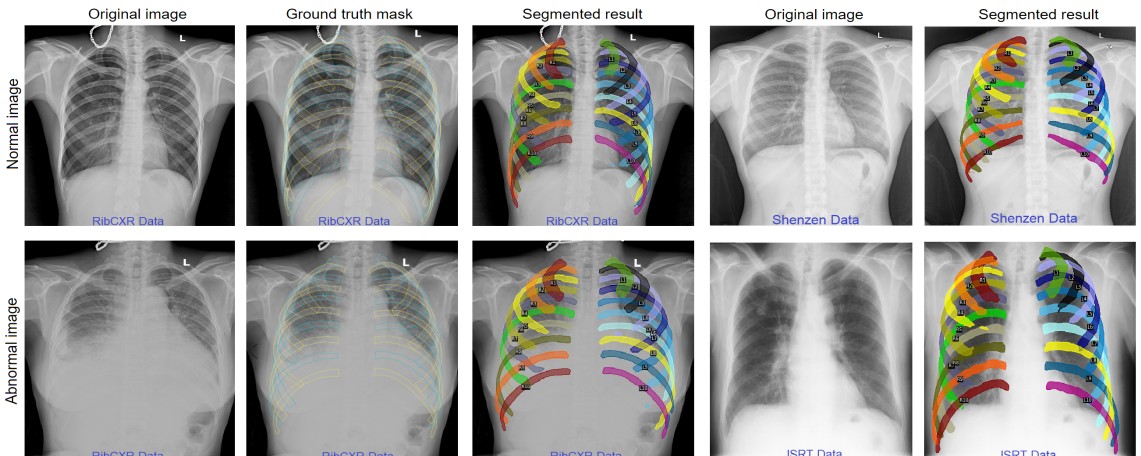

Figure 1: Segmentation results on our validation set and two external datasets.

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
