# OpenReview forum: "VinDr-RibCXR: A Benchmark Dataset for Automatic Segmentation and Labeling of Individual Ribs on Chest X-rays"
_MIDL.io/2021/Conference/Short — MIDL 2021 Poster_

### Official Review · Reviewer_jmBf · 2021-05-01

**Confidence:** 3
**Final Rating:** 3

**Summary:**

This paper comes up with a new Chest X-Ray dataset with instance segmentation annotations which are built for rib segmentation. The dataset has been evaluated on four different network architectures used for semantic segmentation while achieving satisfactory quantitative performance in terms of Dice score and 95% Hausdorff distance.

**Strengths:**

1. The dataset involves instance-based annotations which would be valuable for the community. In addition, the authors commit themselves to open-source the dataset and codes for reproducibility.
2. Novel backbone architectures are also used within the semantic segmentation frameworks.
3. The paper is written clearly with no substantial formatting errors.

**Weaknesses:**

1. Although the dataset provides instance-based segmentation annotations, the experiments were performed only on the semantic segmentation models. Similar to what Wessel et. al. had done, additional experiments for measuring the instance segmentation capabilities of the methodologies would be more intuitive in this sense.
2. Since there are other available datasets for rib segmentation as [1], which has been used in Wessel et. al., it is important to highlight your motivation and identify the differences between your dataset with the others. A comparison between your dataset and [1] can be helpful in that sense.

[1] --> Jens von Berg, Stewart Young, Heike Carolus, Robin Wolz, Axel Saalbach, Alberto Hidalgo, Ana Gim´enez, and Tom´as Franquet. A novel bone suppression method that improves lung nodule detection. International journal of computer assisted radiology and surgery, 11(4): 641–655, 2016.

**Deanonymize Review:**

no

**Detailed Comments:**

1. Please mention what some of the acronyms stand for, e.g. AP/PA.

2. It would be informative to mention the dice overlap between the GT mask and predicted segmentations on the segmented result.

3. Do your models perform segmentation based on a rib/background basis, or does it work like L1/L2/.../R10/BG? In other words, do they perform foreground/background segmentation or multi-label segmentation?

**Justification Of The Rating:**

Despite the flaws of the paper, the dataset looks appealing to benchmark a variety of semantic and instance segmentation methods. In that sense, I believe that the dataset would be interesting for our community.

**Paper Type:**

both

**Special Issue:**

no

---

### Meta-Review · Area_Chair_iwxd · 2021-05-10

**Recommendation:** Accept (Poster)
**Confidence:** 4

**Metareview:**

The reviewer praises the creation of this potentially very valuable dataset and while there are some shortcomings in the presented benchmark/baseline results, I believe these will be quickly addressed by the community if the data becomes more widely used. MIDL aims to give some space for the presentation of open source data and I therefore also recommend acceptance.

---

### Decision · Program_Chairs · 2021-05-11

Accept (Poster)